Research




**Subject Areas:**
ecology, taxonomy and systematics, genomics

**Keywords:**
Ramari's beaked whale, True's beaked whale, *Mesoplodon mirus*, *Mesoplodon eueu*, taxonomy, biodiversity

**Author for correspondence:**
Emma L. Carroll
e-mail: carrollemz@gmail.com



# Speciation in the deep: genomics and morphology reveal a new species of beaked whale *Mesoplodon eueu*

Emma L. Carroll[1], Michael R. McGowen[2], Morgan L. McCarthy[3], Felix G. Marx[4,5], Natacha Aguilar[6], Merel L. Dalebout[7], Sascha Dreyer[3], Oscar E. Gaggiotti[8], Sabine S. Hansen[3], Anton van Helden, Aubrie B. Onoufriou[6,8], Robin W. Baird[9,10], C. Scott Baker[11], Simon Berrow[12], Danielle Cholewiak[13], Diane Claridge[14], Rochelle Constantine[1], Nicholas J. Davison[15], Catarina Eira[16,17], R. Ewan Fordyce[5,4], John Gatesy[18], G. J. Greg Hofmeyr[19,20], Vidal Martín[21], James G. Mead[2], Antonio A. Mignucci-Giannoni[22,23], Phillip A. Morin[24], Cristel Reyes[8], Emer Rogan[25], Massimiliano Rosso[26], Mónica A. Silva[27], Mark S. Springer[28], Debbie Steel[11] and Morten Tange Olsen[3]

[1]School of Biological Sciences Te Kura Mātauranga Koiora, University of Auckland Waipapa Taumata Rau, Auckland 1010, Aotearoa New Zealand
[2]Department of Vertebrate Zoology, Smithsonian National Museum of Natural History, Washington, DC 20560, USA
[3]Section for Evolutionary Genomics, GLOBE Institute, University of Copenhagen, Øster Farimagsgade 5, Copenhagen K DK-1353, Denmark
[4]Museum of New Zealand Te Papa Tongarewa, Wellington, Aotearoa New Zealand
[5]Department of Geology, University of Otago, Dunedin, Aotearoa New Zealand
[6]BIOECOMAC, Department of Animal Biology, Edaphology and Geology, University of La Laguna, San Cristóbal de La Laguna, Tenerife, Canary Islands, Spain
[7]School of Biological, Earth, and Environmental Sciences, University of New South Wales, Kensington 2052, Australia
[8]School of Biology, University of St Andrews, St Andrews KY16 8LB, UK
[9]Cascadia Research Collective, 218 1/2 W. 4th Avenue, Olympia, WA 98501, USA
[10]Hawai'i Institute of Marine Biology, University of Hawai'i, Kaneohe, HI 96744, USA
[11]Marine Mammal Institute and Department of Fisheries and Wildlife, Hatfield Marine Science Center, Oregon State University, Newport, OR 97365, USA
[12]Irish Whale and Dolphin Group, Merchants Quay, Kilrush, Co Clare, Ireland/Marine and Freshwater Research Centre, Galway-Mayo Institute of Technology, Dublin Road, Galway, Ireland
[13]Northeast Fisheries Science Center, National Marine Fisheries Service, National Oceanographic and Atmospheric Administration (NOAA), 166 Waters Street, Woods Hole, MA 02543, USA
[14]Bahamas Marine Mammal Research Organisation (BMMRO), Sandy Point, Abaco, Bahamas
[15]Scottish Marine Animal Stranding Scheme, Institute of Biodiversity, Animal Health and Comparative Medicine, University of Glasgow, Glasgow G12 8QQ, UK
[16]Departamento de Biologia, CESAM and ECOMARE, Universidade de Aveiro, Campus Universitário de Santiago, Aveiro 3810-193, Portugal
[17]Sociedade Portuguesa de Vida Selvagem, Estação de Campo de Quiaios, Rua das Matas nacionais, Figueira da Foz 3080-530, Portugal
[18]Division of Vertebrate Zoology, American Museum of Natural History, Central Park West at 79th Street, New York, NY 10024, USA
[19]Port Elizabeth Museum at Bayworld, Gqeberha 6013, South Africa
[20]Department of Zoology, Institute for Coastal and Marine Research, Nelson Mandela University, Gqeberha 6031, South Africa
[21]Study of the Cetaceans in the Canary Archipelago (SECAC) Casa de Los Arroyo, Arrecife de Lanzarote, Canary Islands, Spain
[22]Caribbean Manatee Conservation Center, Inter American University of Puerto Rico, 500 Carretera Dr John Will Harris, Bayamón 00957, Puerto Rico
[23]Center for Conservation Medicine and Ecosystem Health, Ross University School of Veterinary Medicine, PO Box 334, Basseterre, St Kitts

[24]Southwest Fisheries Science Center, National Marine Fisheries Service, NOAA, 8901 La Jolla Shores Dr., La Jolla, CA 92037, USA
[25]School of Biological, Earth and Environmental Sciences, University College Cork, Ireland
[26]CIMA Research Foundation, Via Magliotto 2, Savona 17100, Italy
[27]Okeanos—Instituto de Investigação em Ciências do Mar & IMAR—Instituto do MAR, Universidade dos Açores, Horta, Portugal
[28]Department of Evolution, Ecology, and Organismal Biology, University of California, Riverside, CA 92521, USA

ELC, 0000-0003-3193-7288; MLM, 0000-0002-9695-6060; OEG, 0000-0003-1827-1493; ABO, 0000-0002-4605-1896; DCh, 0000-0002-0281-1021; RC, 0000-0003-3260-539X; MAS, 0000-0002-2683-309X

The deep sea has been described as the last major ecological frontier, as much of its biodiversity is yet to be discovered and described. Beaked whales (ziphiids) are among the most visible inhabitants of the deep sea, due to their large size and worldwide distribution, and their taxonomic diversity and much about their natural history remain poorly understood. We combine genomic and morphometric analyses to reveal a new Southern Hemisphere ziphiid species, Ramari's beaked whale, *Mesoplodon eueu*, whose name is linked to the Indigenous peoples of the lands from which the species holotype and paratypes were recovered. Mitogenome and ddRAD-derived phylogenies demonstrate reciprocally monophyletic divergence between *M. eueu* and True's beaked whale (*M. mirus*) from the North Atlantic, with which it was previously subsumed. Morphometric analyses of skulls also distinguish the two species. A time-calibrated mitogenome phylogeny and analysis of two nuclear genomes indicate divergence began *circa* 2 million years ago (Ma), with geneflow ceasing 0.35–0.55 Ma. This is an example of how deep sea biodiversity can be unravelled through increasing international collaboration and genome sequencing of archival specimens. Our consultation and involvement with Indigenous peoples offers a model for broadening the cultural scope of the scientific naming process.

# 1. Introduction

The Earth's deep ocean remains less understood than the surface of Mars [1]. However, much biodiversity is waiting to be discovered in the deep sea, and there is great potential for this region to contribute to and challenge major ecological hypotheses [2]. Here, we focus on beaked whales (ziphiids), which are among the most visible inhabitants, due to their large size, worldwide distribution and surfacing to breathe [3]. Even so, their diversity and ecology remain obscure, with seven of the 23 species in the IUCN Red List classified as Data Deficient.

Our understanding of beaked whales has been limited by the scarcity of records, with many species known only by a handful of incomplete skeletons [4]. Beaked whales typically spend limited time at the surface [5], during which they are difficult to distinguish [6]. Genetic tools have been instrumental in developing an understanding of the diversity and phylogenetic relationships among ziphiids [7], but the application of genomic tools to understand their ecology and evolution has been limited [8] despite recent technological advances [9].

Here, we use a range of genomic approaches to access genetic information from museum and archival specimens of varying age and quality to investigate the taxonomic status of disjunct populations of True's beaked whale (*Mesoplodon mirus*). First described by Frederick W. True in 1913 from a male that stranded in North Carolina, USA [10], the known distribution (figure 1) of *M. mirus* grew to include much of the temperate North Atlantic (NA) [3,6,12]. In 1959, the species was first reported in the Southern Hemisphere (SH) off South Africa [13], with the subsequent discovery of a breeding population [14]. Additional SH specimens were reported from Mozambique [15], Tristan da Cunha [16], Walter Shoals (south of Madagascar) [3], southern Brazil [17], Australia [18] and Aotearoa New Zealand [11] (figure 1).

NA and SH True's beaked whales are separated by thousands of kilometres, raising questions about their level of divergence [6,19]. Initial suspected differences in pigmentation between hemispheres [19] have been outweighed by greater variation in colouration within NA, reducing evidence for differentiation [6]. Similarly, the only comprehensive anatomical analysis to date (based on six SH and nine NA specimens) found differences in just one cranial measurement out of 47 [19]. Molecular analyses did reveal notable divergence between NA and SH specimens, but this was based on mitochondrial control region sequences from four individuals [20,21] and *ad hoc* data collected thereafter for species identification [6,11]. Here, we take an integrative approach, combining genomic and morphological datasets, to resolve this taxonomic quandry.

# 2. Results

## (a) Genomes reveal deep genetic divergence

To resolve the status of SH 'True's' beaked whales, we first examined interhemispheric divergence using both complete mitogenomes and nuclear single-nucleotide polymorphisms (SNPs; figure 2). A Bayesian phylogeny of ten NA and two SH mitochondrial genomes clustered individuals from each hemisphere into well-supported, reciprocally monophyletic sister clades. Within this phylogeny, the holotype of *M. mirus* (USNM175019) clearly groups within the NA group (figure 2*c*); the holotype for the SH form clusters with the other SH sample with high support. The same pattern appears in a phylogeny constructed using 17 217 SNPs derived from ddRADseq for five samples from each hemisphere and three other beaked whale species (44 total samples; figure 2*a*; electronic supplementary material table S1 and dataset S1). The divergence between *M. mirus* in NA and SH appears greater than in the two other globally distributed beaked whales: Cuvier's beaked whale (*Ziphius cavirostris*, $n = 15$) and Blainville's beaked whale (*M. densirostris*, $n = 9$) (figure 2*a*). The ddRAD genetic clustering results revealed little admixture between the NA and SH samples, suggesting no major introgression between the two forms after divergence (figure 2*b*).

The deep genetic split between the NA and SH is supported by high levels of genetic differentiation based on 15 761 ddRADseq SNPs in *M. mirus* (NA/SH $n = 5/5$, $F_{ST} = 0.64$, $p < 0.01$, with 1909 (12%) fixed differences, estimated error rate of 0.002 per SNP allele), mitogenomes (NA/SH $n = 9/2$, $d_A = 0.04$, $F_{ST} = 0.96$, $p < 0.01$) and a shorter fragment of the mitochondrial control region for which more samples were available (304 bp, NA/SH $n = 19/14$, $d_A = 0.04$, $F_{ST} =$

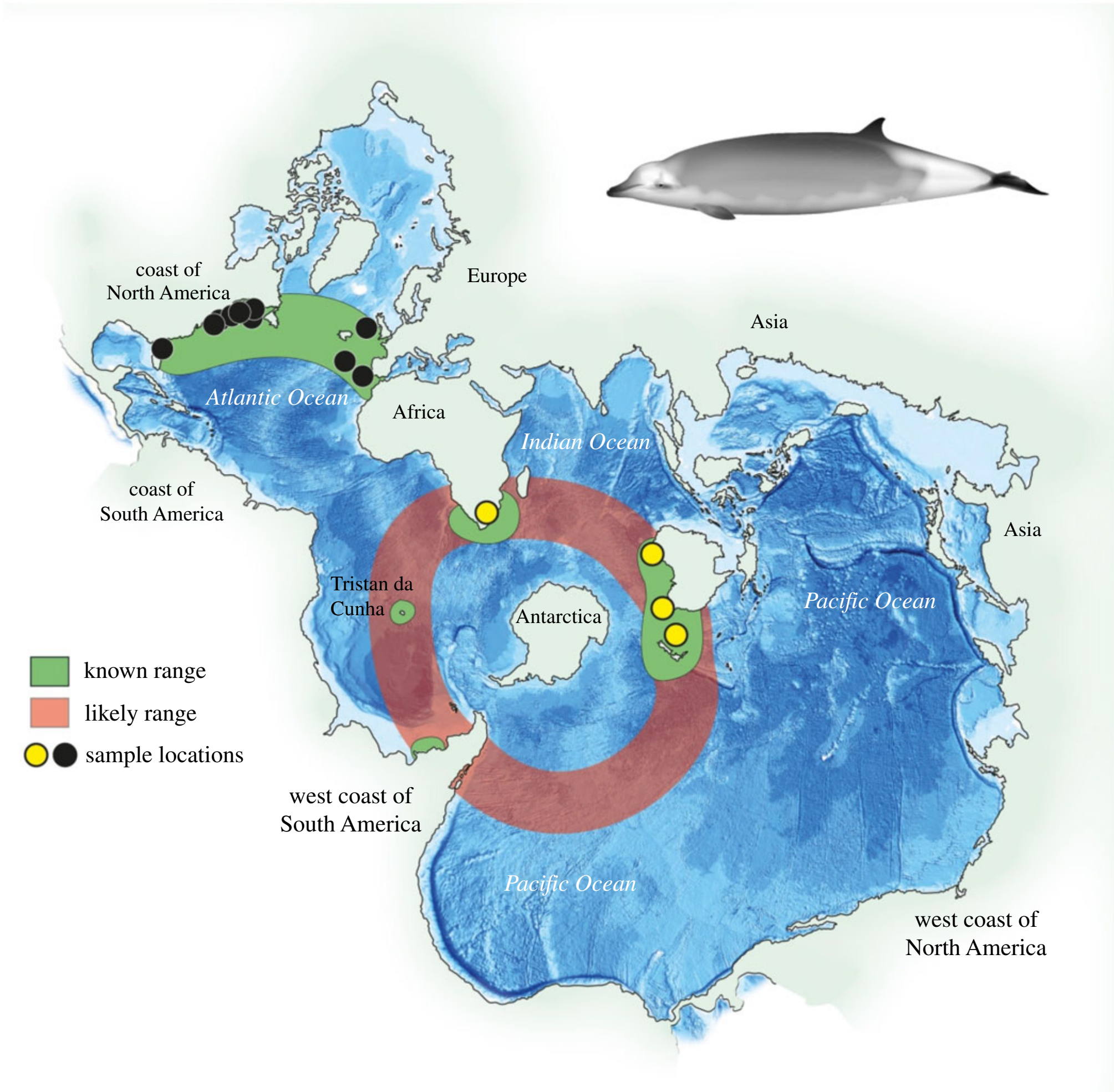


**Figure 1.** Sampling locations in the NA (black circles) and SH (yellow circle). Global map viewed as a Spilhaus projection that shows the connectedness of the ocean, with sampling locations and distribution of *Mesoplodon mirus* and proposed species *M. eueu* shown by the key (sourced from [3,11]), with the artist's impression of the species in top right. Credit: Vivian Ward, University of Auckland.

0.85, $p < 0.01$; electronic supplementary material, table S2). This level of differentiation is consistent with that proposed to indicate species separation in cetaceans [23].

A time-calibrated Bayesian analysis of the mitochondrial genomes of the focal species and its closest relative based on previous cetacean phylogenies [24] dates the NA/SH split to approximately 2 Ma (95% HPD: 1.4–2.6 Ma; figure 2*c*). To examine the pattern of divergence further, we sequenced two whole nuclear genomes using Illumina short-read technology, one from each ocean basin (NA, 51x and SH 7x coverage). An associated hybrid pairwise sequential Markovian coalescent (hPSMC) analysis suggests that the cessation of gene flow likely occurred around 0.55–0.35 Ma (figure 2*d*).

The SH dataset has significantly higher nuclear diversity (SNP observed heterozygosity $H_O = 0.159$, 95% CI: 0.156–0.163, *t*-test $t = -10.85$, Kolmogorov–Smirnov $D = 0.053$, $p < 0.001$, genome-wide heterozygosity $h = 0.00223$, 95% CI: 0.00220, 0.00225, $t = 35.49$, $p < 0.001$) than the NA dataset ($H_O = 0.134$, 95% CI:0.131–0.1370, $h = 0.00165$, 95% CI: 0.00162, 0.00168), but there was no significant difference in mitochondrial control region haplotype diversity ($H_d$, 304 bp) between the two regions (SH $H_d = 0.84$, 95% CI: 0.813–0.867, NA $H_d = 0.80$, 95% CI:0.772–0.828, permutation test [25], $p > 0.05$). This is consistent with the estimates of effective population size during the last glacial maximum derived from whole-genome data using PSMC of approximately 17 000 for the SH and approximately 13 000 from NA (figure 3). Although down-sampling flattened the PSMC trend (electronic supplementary material, figure S1), the recent expansion was consistent with the Tajima's D statistic from ddRAD data suggesting that both the NA and SH forms have undergone expansion in recent evolutionary time (SH = −0.11, 95% CI: −0.134, −0.089, $t = -9.82$; Wilcoxon signed-rank test $V = 14240039$, NA = −0.164, 95% CI: −0.187, −0.141, $t = -13.856$, $V = 10858056$, $p < 0.001$ for all analyses).

## (b) The skulls of the two forms are distinct

To determine morphological distinctiveness, we conducted a geometric morphometric analysis of 15 NA and 23 SH skulls, including the holotype of *M. mirus* and holotype of the SH form (figure 4*a–e*; electronic supplementary material, dataset S2). A principal component analysis (PCA) and associated PERMANOVA ($F = 18.98$; $p = 0.0001$) of eight size-standardized cranial measurements (electronic supplementary material, dataset S2, figure S2) clearly separate NA and SH individuals (figure 4*f*; electronic supplementary material, figure S3). The first two principal components account for 83.3% of variance and reveal the southern form has a shorter rostrum and mandibular symphysis (PC1: 56.25%), a broader rostral base and slightly more expanded premaxillary crests and sac fossae (PC2: 26.31%; first six PCs comprise 96% of the variation).

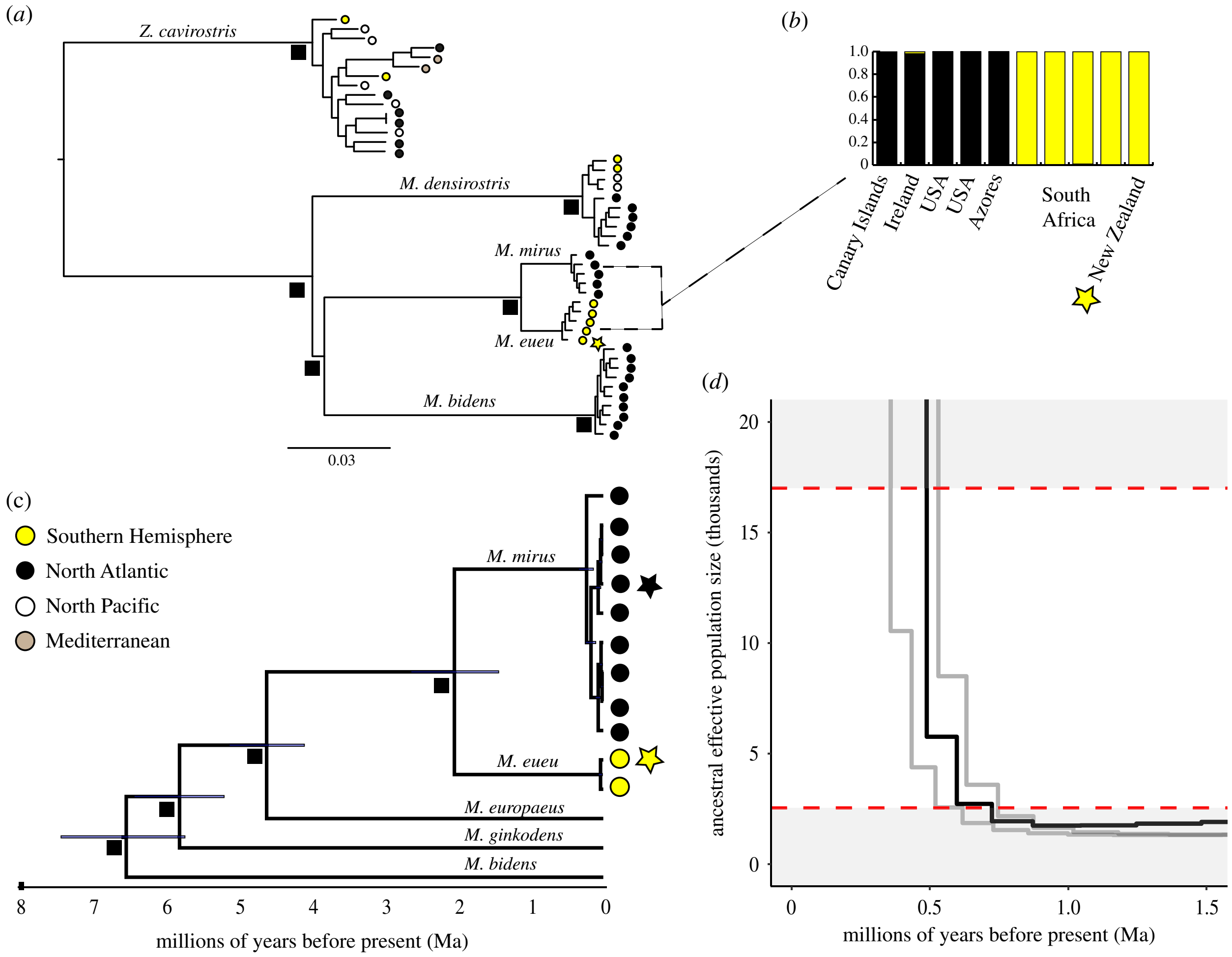


**Figure 2.** Deep genetic divergence between *M. mirus* (black) and *M. eueu* (yellow). (*a*) ddRAD-based phylogenetic tree on 17 217 SNPs, with 95% bootstrap support indicated by black boxes; (*b*) inset showing ddRAD-based admixture plot of *M. mirus* and *M. eueu*, based on 15 761 SNPs variable in these two species. (*c*) Mitogenome phylogenetic tree with 95% HPDs indicated by black boxes. (*d*) hPSMC analysis of nuclear genomes from one specimen each from USA and South Africa showing the end of gene flow between the Northern Hemisphere and SHs estimated at 0.35–0.55 Ma. The black line is the hPSMC result from our observed data and the grey lines represent the simulations closest to the real data without overlapping it, following Li and Durbin [22]. Holotype samples in (*a*) and (*c*) indicated with yellow and black stars.

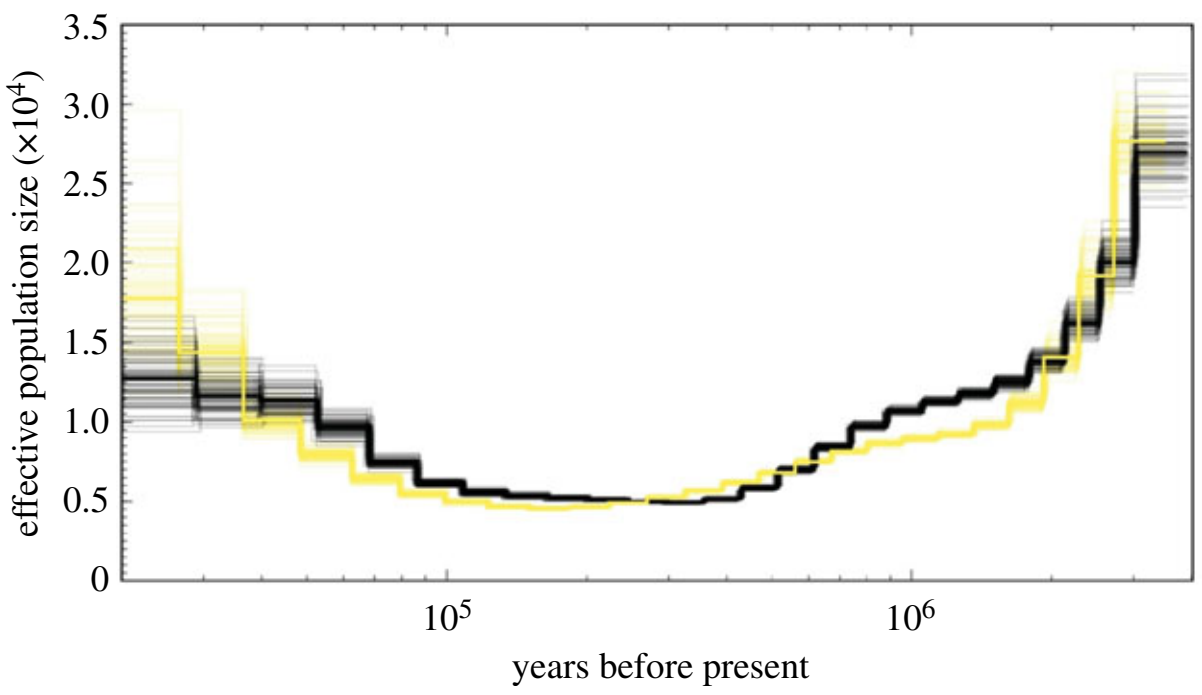


**Figure 3.** Historical effective population sizes of *M. mirus* (black) and *M. eueu* (yellow) over time. Pairwise sequential Markovian coalescent reconstruction of the two species effective population sizes from 20 000 to 3 million years before present, assuming a generation span of 25.9 years [26] and a mutation rate [27,28] of $2.3 \times 10^{-8}$. The dark single line represents the median and the lighter lines represent 100 bootstrap.

Both visual examination and morphometric analysis ($F = 3.93$, $p = 0.017$) show that the SH individuals are sexually dimorphic (female: figure 4; male: electronic supplementary material, figures S4–S6). Females (figure 4) have a longer mandibular symphysis, while males (electronic supplementary material, figure S4) are significantly larger (greater bizygomatic width: mean = 359.4 mm versus 346.6 mm; $t = 2.39$, $p = 0.028$) and have a pair of large, erupted apical 'tusks', more ossification of the mesorostral cartilage (electronic supplementary material, figure S6 [19]), a wider rostrum base, and broader premaxillary crests and sac fossae. In the NA form (female: electronic supplementary material, figure S7; male: electronic supplementary material, figure S8), males have enlarged 'tusks', more mesorostral ossification, broader premaxillary crests and a shorter mandibular symphysis also ($F = 4.98$, $p = 0.013$; electronic supplementary material, figure S3) but the sexes are comparably sized (mean bizygomatic width = 350.25 mm versus 348.43 mm; $t = 0.34$, $p = 0.74$).

## (c) Introducing a new species: Ramari's beaked whale, *Mesoplodon eueu* sp. nov

Molecular and morphological data reveal that True's beaked whales from the NA and SH form two distinct, long-divergent evolutionary lineages, consistent with species under the Genealogical Concordance Species Concept [29]. Therefore, we propose that the SH form should be reclassified as a new species.

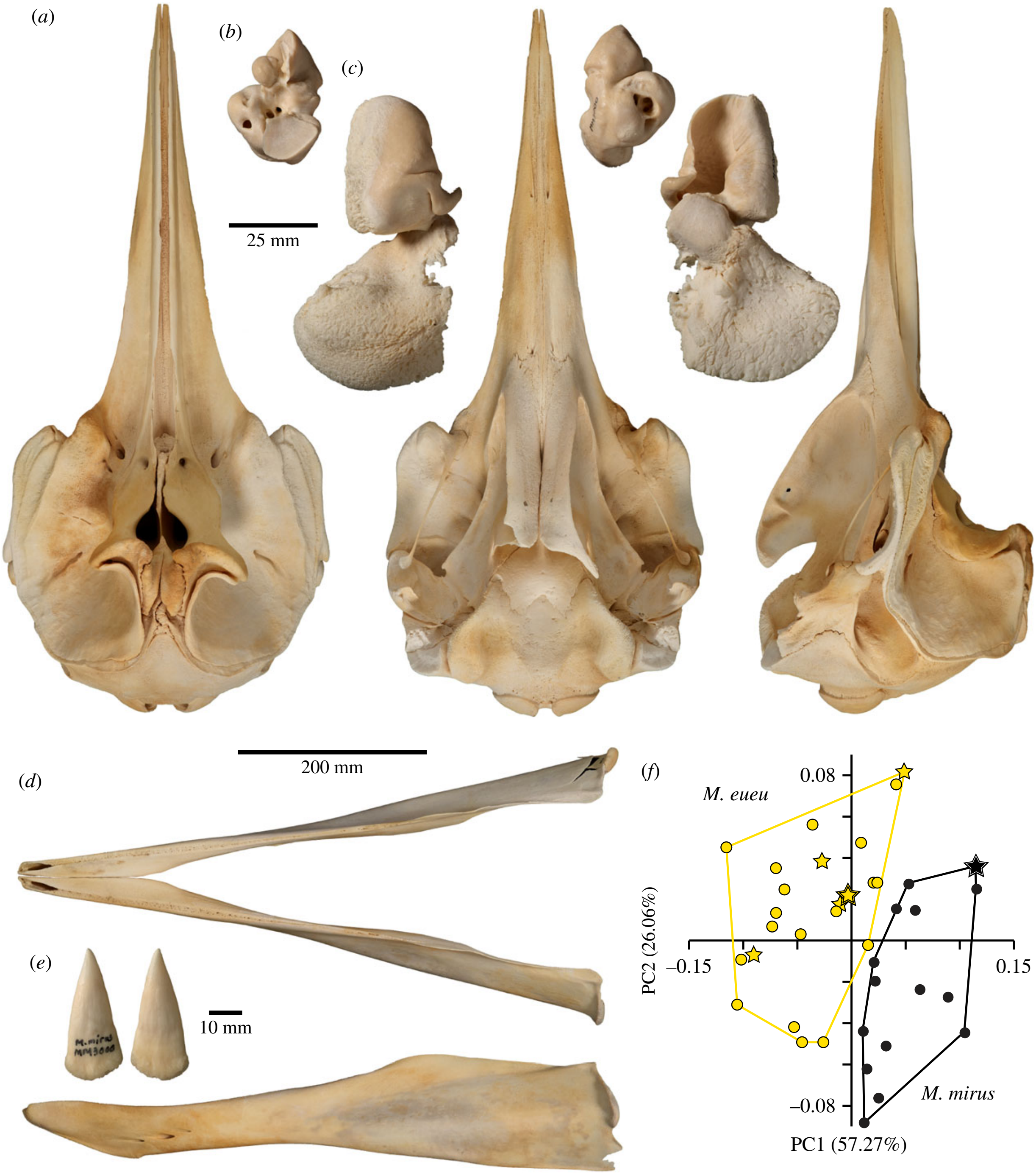


**Figure 4.** Skull and morphological distinctiveness of *M. eueu* shown by holotype (NMNZ MM003000). (*a*) Skull in dorsal (left), ventral (centre) and left lateral (right) view (*b*) periotic in dorsal (left) and ventral (right) view; (*c*) tympanic bulla in dorsal (left) and ventral (right) view; (*d*) mandible in dorsal (top) and lateral (bottom) view; (*e*) mandibular tusks in medial (left) and lateral (right) view. (*f*) PCA of cranial and mandibular measurements showing clear separation between *M. eueu* from the SH and *M. mirus* from the NA. Percentages next to principal components (PCs) denote the total variance explained. Filled stars are holotypes and hollow stars paratypes. (Online version in colour.)

**Systematic biology**
Cetacea Brisson 1762
Ziphiidae Gray 1865
*Mesoplodon* Gervais 1850
*Mesoplodon eueu* sp. nov.
*Holotype*
NMNZ MM003000, a pregnant, 5.06 m long adult female named Nihongore by Te Rūnanga o Makaawhio. Collected by Ramari Stewart, Nathaniel Scott and Don Neale after beach-cast on 27 November 2011. The complete skeletons of the female and fetus are held by Museum of New Zealand Te Papa Tongarewa (NMNZ, Wellington, Aotearoa New Zealand; specimen MM003000), and a tissue sample is held in the New Zealand Cetacean Tissue Archive (NZCeTA, University of Auckland, Auckland, Aotearoa New Zealand; all institutional abbreviations in electronic supplementary material, S1).

*Type locality*
Waiatoto Spit, South Westland, Aotearoa New Zealand.

*Paratypes*
Adult females (PEM N0136 and PEM N3438) and adult male (PEM N1114) held at Port Elizabeth Museum (Gqeberha, South Africa) and adult males (SAM-ZM-041596 and SAM-ZM-039840) held at Iziko South African Museum (Cape Town, South Africa). Full description of paratypes is found in electronic supplementary material S2.

*Etymology*

The scientific and common names acknowledge links with Indigenous communities in South Africa and Aotearoa New Zealand, respectively, and were chosen in consultation with these peoples. Most of the South African strandings come from territory inhabited by the Khoisan peoples. Guided by the Khoisan Council, we chose the name //eu//’eu (simplified to *eueu* to fit nomenclature standards; correct pronunciation available in associated audio clip a1 in the electronic supplementary material), which means ‘big fish’ in Khwedam (from the Khoe language family). In Aotearoa New Zealand, Māori cultural expert Brad Haami developed a shortlist of potential names meaningful in the Māori language, which was then sent for comment to Te Rūnanga o Ngāi Tahu. The selected common name, Ramari’s beaked whale, pays homage to Māori tohunga (expert) Ramari Stewart, who has kept traditional knowledge alive, contributed extensively to scientific research on marine mammals, and helped prepare the skeleton of the holotype. The word ‘Ramari’ means a rare event in the Māori language, reflecting the elusive nature of most beaked whales.

*Diagnosis*

*Molecular characteristics*

*M. eueu* differs from *M. mirus* based on nuclear DNA markers, and from *M. mirus* and its closest relatives *M. europaeus*, *M. ginkgodens* and *M. bidens* using mtDNA markers (figure 2). *M. mirus* is distinct from all other mesoplodont species based on previous mitochondrial and nuclear DNA trees [7,20,24].

*Mitochondrial DNA:* analysis of mitochondrial data includes sequences from the holotypes of both *M. mirus* and *M. eueu* at all sequence lengths. Over the 304 bp mitochondrial control region segment, *M. eueu* is distinguished by seven fixed differences from *M. mirus*, with $F_{ST} = 0.85$ ($p < 0.01$), $d_A = 0.04$ between the two species. At the full mitochondrial genome lengths, *M. eueu* is distinguished by 579 fixed differences from *M. mirus* with $F_{ST} = 0.96$ ($p < 0.01$) and $d_A = 0.04$ (electronic supplementary material, table S2).

*Nuclear DNA:* reduced representation sequencing with ddRAD showed *M. eueu* had a distinct admixture pattern to *M. mirus* (figure 2), and an $F_{ST} = 0.64$ ($p < 0.0001$) was estimated between the two species. *M. mirus* and *M. eueu* were distinguished by 1909 fixed differences (12%, per SNP allele error rate = 0.002), across a dataset of 15 671 SNPs found between or within both species. Comparison of one whole nuclear genome each from *M. mirus* and *M. eueu* showed a level of nucleotide divergence of 0.28%.

*Morphological characters*

*M. eueu* is a larger (5.3 m) species of *Mesoplodon* differing from all other members of the genus except *M. mirus*, *M. hectori* and *M. perrini* in having tusks positioned at the tip of the mandible. It also differs from *M. hectori* and *M. perrini* in having smaller, less triangular tusks and from *M. mirus* in having a relatively shorter rostrum with a wider base, a shorter mandibular symphysis, wider premaxillary sac fossae and crests, and a taller cranium.

*External appearance*

The external appearances of *M. eueu* and *M. mirus* are not known to be consistently distinguishable. Both species are rotund mesoplodonts with bodies that taper towards the tail and rostrum, with somewhat bulbous and well-defined melons, a mostly straight beak and short, straight gape. Colouration is generally grey with a dark eye patch in both species, but there may be specific colouration patterns linked to *M. eueu*; for example, a female that stranded in South Africa showed a whitish dorsal colouration from the fin to caudal peduncle [14]. However, Aguilar *et al.* [6] reported *M. mirus* with varying dorsal and ventral white colouration in the Canary and Azores Islands, so colouration patterns linked to species cannot yet be conclusively defined.

*Distribution*

*M. eueu* probably occurs throughout temperate SH waters, with reports from several locations. Genetic methods have confirmed at least some of these records in South Africa ([20], this study), Australia [20] and Aotearoa New Zealand [11].

*Nomenclatural acts*

This published work and the nomenclatural acts it contains are registered in ZooBank (http://zoobank.org/), the online registration system for the International Commission on Zoological Nomenclature (ICZN). The ZooBank Life Science Identifier is urn:lsid:zoobank.org:pub:C61A1A33-B234-476E-AA24-86C6ED130C10, and for the new species is urn:lsid:zoobank.org:act:C56121C4-1E15-4A07-A270-D92AF43AE74A.

## 3. Discussion

As many as 1.5 million species await discovery in the deep sea [30]. Here, we show that detailed analysis of even a small number of samples can yield profound insights into the diversity and phylogeography of the species that occur in this vast habitat. As one of the few mammalian deep sea specialists, it is perhaps not surprising that ziphiids are among the most speciose cetacean lineages. Our results emphasize this pattern and corroborate the idea that the deep sea is more biodiverse than previously thought [2].

Deep sea ecosystems are governed by temperature, primary productivity and habitat complexity [31]. Modelling of energy input suggests that deep sea biodiversity peaks at latitudes of 30–50° [32], coinciding with the ranges of *M. eueu* and *M. mirus*. Based on our time-calibrated, phylogenetic reconstruction of mitogenomes, the initial divergence of these species may have been driven by intense cooling in tropical ocean temperatures approximately 2 Ma, which in turn would have facilitated cross-equatorial dispersal from one hemisphere into another. It is unclear in which ocean basin the ancestral species originated, but the slightly greater genetic diversity and wider geographical range of *M. eueu* may indicate a southern origin. Either way, our hPSMC analysis suggests that by 0.35 Ma all significant gene flow between the two hemispheres had ceased. After divergence, the population size of both species probably expanded as *M. mirus* and *M. eueu* showed signals of expansion in recent evolutionary time.

Examples of anti-tropical species pairs and population structuring are common among cetaceans and other marine mammals [33,34]. In our analysis, *M. densirostris* also shows distinct populations in the NA and Indo-Pacific (figure 2*a*), but with far less divergence than *M. mirus* and *M. eueu*. This may indicate dispersal during a more recent glacial period, or greater gene flow in *M. densirostris* with its more continuous, tropical distribution [3]. *Mesoplodon eueu* is the fifth beaked whale species to be described or elevated to species status in the past few decades [21,35–37]. Ziphiids occur primarily offshore, spend little time at the surface [5] and are hard to distinguish visually [6], which makes them

difficult to study. An integrative approach combining genetic and morphometric analysis has proved effective at uncovering the diversity and relationship of these elusive animals [21,36], and is likely to produce further taxonomic insights in the future. Our study required international collaboration, generating a global archive (International Tissue Archive for Beaked Whale) of ziphiid samples collected over five decades. Exploiting everything from whole-genome sequences to short mtDNA fragments allowed us to use samples of varying quality and thus maximize geographical coverage. Overall, our results highlight the value of museum and tissue archives in documenting and understanding speciation, especially when paired with techniques like genomic sequencing.

Describing a new species requires important decisions around the naming of the new taxon. We sought to recognize Indigenous peoples' deep connection with, and knowledge of, the natural environment by consulting with them on potential species and common names. This is part of a critical shift in the global science community, which strives to collaborate with Indigenous knowledge holders in ecology [38,39] and conservation biology [40]. Here, this has resulted in one of the first cetaceans named after an Indigenous woman.

## 4. Methods

### (a) Dataset overview

The geographic origin of and analyses used in each sample is summarized by species in electronic supplementary material, table S1. Full metadata, including sex, collection date and location, are given for samples in the genomic and morphometric analyses in the electronic supplementary material, datasets S1 and S2, respectively.

### (b) Investigating genomic differentiation

#### (i) DNA extraction and genomic sequencing

DNA extraction method varied depending on the sample type. For tissue, DNA was extracted either using the Qiagen DNeasy kit, Gentra Puregene kit or using standard phenol:chloroform methods [41,42]. For bone, extraction followed the silica-based method [43], as modified by [44] or [45]. Genomic sequencing was conducted using different methods and platforms for double digest Restriction Associated DNA (ddRAD), shotgun nuclear and mitogenome analyses (see electronic supplementary material, S3). To increase the sample size for the mitochondrial control region analysis, we combined novel unpublished with previously published [6,11,20] data (electronic supplementary material, dataset S1). PCR amplification was conducted with forward primer Dlp1.5 [46] and reverse primer Ha500R_R (5′-CCATCGAGATGTCTTATTTAAGAGG-3′) using standard protocols [7]. PCR products were sent to Macrogen Europe for Sanger sequencing.

#### (ii) ddRAD data analysis

STACKS v. 2 [47] was used to genotype the samples using the reference mapped pipeline, employing the algorithm of Maruki & Lynch [48] that accounts for sequencing quality, allele balance and allele frequencies. For the phylogenetic analysis, we used a dataset comprising the species of interest, *M. mirus*, two other globally distributed beaked whales to compare divergence patterns, *M. densirostris* and *Ziphius cavirostris*, and the more closely related *M. bidens* (electronic supplementary material, table S1). For the population genomics analysis, we focused on *M. mirus* samples. Both analyses followed the same pipeline. First, reads were demultiplexed using the process_radtags command in the STACKSv2 program and aligned against the *M. bidens* genome (GenBank: PRJNA399476) using BWA-MEM v. 0.7.17 [49]. The SNP and genotyping calling likelihood $\alpha = 0.01$, ensuring only high-quality variants were called. We used a tiered and iterative approach to filtering, starting with low cut-off values for missing data (applied separately per locus and individual) and finalizing the dataset with higher thresholds, an approach that can maximize available data [50]. Genotyping error rate and discordant loci were identified by running one sample through the laboratory work and genotyping pipeline twice. Differences between these replicates were used to identify discordant loci that were then removed from the analysis and estimate a per SNP allele error rate for the pipeline. This QC pipeline was done in R v. 3.6.0 [51] and VCFtools v. 0.1.12a [52] and code is available at https://github.com/emmcarr/Mmirus.

Loci were exported from STACKS v2 as a phylip file and a total of 14 468 parsimony informative sites were used to construct a maximum-likelihood phylogenetic tree in IQtree [53] using a general time-reversible model with unequal rates and unequal base frequencies [54] and ascertainment bias correction. No partitioning was used. Confidence in the clades was given by standard nonparametric bootstrapping ($n = 1000$).

We estimated levels of genetic differentiation using $F_{ST}$ calculated with the R package hierfstat [55], with significance assessed using the permutation method in the R package strataG [56]. The number of fixed differences between the NA and SH was estimated using the R package dartR [57] and Tajima's D was estimated for each region using vcftools (sliding window of 10 000 bases), with significance from zero of the latter tested with t-test and Wilcoxon signed-rank test in R. We estimated mean and 95% CI for observed heterozygosity [58] for NA and SH using hierfstat and compared them using *t*-test and Kolmogorov–Smirnov tests in R. Admixture was assessed using the genetic clustering program sNMF [59] implemented in the R package LEA [60]. The best *k* was inferred by calculating cross-entropy values from 10 runs of *k* set from 1 to 4.

#### (iii) Mitochondrial genome (mitogenome) assembly

Mitochondrial reads were assessed prior to and after trimming and filtering reads with FastQC v. 0.11.8a [61]. Adapters were removed and sequences were trimmed in BBDUK (options: ref = adapters ktrim = r $k = 23$ min $k = 8$ h dist = 1 tbo qtrim = rl trimq = 15 ma $q = 20$ min len = 40: https://jgi.doe.gov/data-and-tools/bbtools/bb-tools-user-guide/bbduk-guide/). Trimmed reads were mapped to a *M. mirus* reference mitogenome (Genbank: NC_042217.1) using BWA-MEM. PCR duplicates were removed with SAMtools v. 1.9 (rmdup command: [62]). The consensus sequence was generated with ANGSD v. 0.931 [63] (options: -doFasta2 -doCounts1 -minQ30 -minMapQ30 -setMinDepth3).

#### (iv) Bayesian mitogenome phylogeny

All mitogenomes were aligned using MAFFT v. 7.388 [64] with *M. bidens*, *M. ginkgodens* and *M. europaeus* as outgroups, *M. mirus*'s most closely related species [24]. The 12 s rRNA, 16 s rRNA and 13 protein-coding mitochondrial genes were extracted and individually aligned. Stop codons in coding genes were manually removed in Geneious Prime 2020.1.2 (https://www.geneious.com/). The final alignments were concatenated into a combined alignment and substitution models were inferred with PartitionFinder v. 2.1.1 [65]. There were eight subsets (electronic supplementary material, table S3) which were adjusted to fit the BEAUTi v. 2.5.2 [66] XML file generator and linked with a relaxed clock log normal and a linked Yule tree model. Three MRCA node priors were implemented based on fossil evidence [24] (electronic supplementary material, table S4). The MCMC model was implemented in BEAST2 v. 2.5.2 [66] with 15 million chains and

sampled every 1500 steps. The model was run three separate times to ensure convergence. The log files were inspected with Tracer v. 1.7.1 (https://github.com/beast-dev/tracer/releases/tag/v1.7.1) and combined with LogCombiner v. 2.5.2 after a 10% burn-in. The tree files were combined after a 10% burn-in with LogCombiner and a Maximum Clade Credibility (MCC) tree was generated with TreeAnnotator v. 2.4.2 [66]. The final MCC tree was visualized in FigTree v. 1.4.4 (http://tree.bio.ed.ac.uk/software/figtree/).

#### (v) mtDNA control region analysis

The mtDNA control region sequences were aligned and trimmed using Geneious Prime 2020.1.2. Given their origins, sequences varied in length and overlap. We created three datasets of lengths 600 bp ($n = 28$), 423 bp ($n = 32$) and 304 bp ($n = 33$), balancing length and sampling representation. The phylogenetic analyses were conducted on each datasets using MrBayes [67] (GTR gamma: 1 000 000 replicates, 100 000 burn-in) and NJ (HKY: 10 000 bootstraps) as implemented in Geneious Prime. Haplotype and nucleotide diversity, as well as estimates of $d_A$, $F_{ST}$, $K_{ST}$ and number of fixed differences between ocean basins were estimated in DNAsp v. 6 [68]. Significant differences in diversity statistics between NA and SH were assessed using a permutation test [25].

#### (vi) Nuclear genome assembly and haploid consensus sequence generation

Genomes were assembled following steps modified from [69]. For genome assembly, raw sequences were inspected for QC with FastQC. Sequence adapters and short reads were trimmed using BBDUK and subsequently re-assessed with FastQC. Remaining reads were mapped to the *M. bidens* [8] reference genome using the BWA-MEM algorithm. PCR duplicates were removed with Samtools -rmdup command. Consensus haploid sequences for hPSMC were generated with ANGSD (minimum quality score = 25, minimum mapping score = 5, minimum read depth = 3, -uniqueonly 1 and -remove_bads 1). Autosomal scaffolds and scaffolds shorter than 100 000 base pairs were filtered out using the filterbyname.sh script from BBMAP v. 28.70 (https://jgi.doe.gov/data-and-tools/bbtools/bb-tools-user-guide/bbmap-guide/). Assembly depth of coverage estimates were generated with SAMtools. The NA assembly was downsampled to the same depth of coverage of SH assembly using SAMtools to account for inflated levels of heterozygosity in the lower depth of coverage assemblies. Subsequently, the two genome heterozygosity estimates were calculated with ROHan [70]. Windows containing less than 90% called sites were filtered out, the mean and 95% CI were calculated each for NA and SH, and the distributions compared with a *t*-test in R.

#### (vii) Pairwise sequential Markovian coalescent and hybrid PSMC modelling

The PSMC plot was generated following [22]. Sex-linked scaffolds [8] were removed from the assembly and a subsequent diploid genome was generated with bcftools using a minimum depth of coverage of five and mapping quality score of 30. The depth of coverage for *M. mirus* was downsampled to 5x–40x using SAMtools for a more similar comparison and to see the effects of the depth of coverage on PSMC plots (electronic supplementary material, figure S1). The PSMC file was generated and visualized in gnuplot v. 5.2 (options: –N25-t15-r5-p"4 + 25 * 2 + 4 + 6": http://www.gnuplot.info/). The plot was scaled using a general odontocete mutation rate of $2.34 \times 10^{-8}$ per year [27], following previous similar work [28], and a generation time of 25.9 years (based on *Berardius* [26]).

The two nuclear consensus genomes were combined into a single pseudo-hybrid psmcfa file with the psmcfa_from_2_fastas.py script [71]. A PSMC run was implemented as above [22] with a modified -s10 setting. From the hybrid *M. mirus* hPSMC output, we manually visualized the text file output from the psmc_plot.pl script to estimate the pre-divergence $N_e$. The $N_e$ of 17 000 was incorporated to run simulations dating the end of gene flow from 300 000 to 600 000ya in 50 000 year intervals using ms [72]. All simulations were plotted with the real data and the range for dating the end of gene flow was estimated from the two simulations which occurred closest to the real data without overlapping it, within the range of 1.5–10 × the pre-divergence $N_e$.

#### (viii) Identity by state calculation

The haploid pairwise identity between NA and SH sample partitions was calculated in ANGSD. The identity by state (IBS) value was calculated with the -doIBS 2 (consensus base) command and included filters including only sites present in both bam files with a minimum depth of coverage of 5 and reads that had a mapping score of 30, reads that mapped uniquely (samtools flag 256) and bases with a quality score of 30.

### (c) Investigating morphological variation

#### (i) Skull morphometrics

We quantified the morphology 15 NA and 23 SH specimens via eight measurements of the cranium and mandible (electronic supplementary material, figure S5). Our sample included the holotype of *M. mirus* (USNM175019) and *M. eueu* (NMNZMM003000). All measurements were taken to the nearest mm, divided by the bizygomatic width of the skull to account for differences in body size (see [73] for the use of this metric as a size proxy in cetaceans), log-transformed and summarized via PCA using PAST 4.03 [74]. We tested for (i) interhemispheric differences and (ii) sexual dimorphism per hemisphere via a PERMANOVA. Missing values were estimated via mean value imputation (PCA) or accounted for via pairwise deletion (PERMANOVA: both PAST defaults).

Ethics. Biopsy samples were collected from *M. mirus* in the NA under NMFS MMPA Permit #21371 and animal ethics protocol IACUC-2015-007 to D.C. All other tissue samples come from beach cast animals that stranded between 1977 and 2018 across the species' global distribution (figure 1; electronic supplementary material, dataset S1).

Data accessibility. Code is available on the Github respository: https://github.com/emmcarr/Mmirus. Metadata, including accession IDs, for genomic samples are in the electronic supplementary material [75]. Data are archived on NCBI (Bioprojects PRJNA766520 for whole-genome data and PRJNA765474 for ddRAD data), European Nucleotide Archive (Accession ID PRJEB47691 with details in electronic supplementary material, dataset S1) and Dryad (https://doi.org/10.5061/dryad.wpzgmsbnq: ddRAD raw and demultiplexed data, mtDNA alignments at all sequence lengths, complete mitogenomes that are also available on NCBI).

Authors' contributions. E.L.C., M.R.M., M.T.O.: conceptualization, data curation, formal analysis, funding acquisition, investigation, methodology, project administration, resources, visualization, writing—original draft, writing—review and editing; M.L.M., F.G.M., G.J.G.H.: data curation, formal analysis, investigation, methodology, resources, visualisation, project administration, writing—original draft, writing—review and editing; M.L.D.: conceptualization, data curation, funding acquisition, investigation, methodology, project administration, resources, writing—original draft, writing—review and editing; O.E.G.: funding acquisition, investigation, writing—original draft, writing—review and editing; A.v.H., P.A.M., S.D., S.S.H., A.B.O., D.S., C.R.: data curation, formal analysis, resources, investigation, writing—original draft, writing—review and editing; R.C., project administration, funding acquisition, resources, writing—original draft, writing—review and editing R.B., N.A.S., C.S.B., S.B., D.Ch., D.Cl., N.J.D., C.E., R.E.F., J.G., V.M., J.G.M., A.A.M.-G., E.R., M.R., M.A.S., M.S.S.: funding acquisition, resources, writing—original draft, writing—review and editing.

Competing interests. The authors declare no competing interests.

Funding. This work was supported by ONR grants N000141613017 to E.L.C. and N.A. and N00014-18-1-2808 to C.S.B.; funds from the NMNH Rebecca G. Mead and James G. Mead Marine Mammal

Endowment, NSF (USA) grant no. DEB-1457735 to M.S.S., P.A.M. and J.G.; Brothers Hartmann Foundation grant no. AB28148 to M.T.O.; NMFS, BOEM, and USA Navy funding to D.Ch. under the Atlantic Marine Assessment Program for Protected Species. M.L.M. was funded under the Marie Skłodowska-Curie grant agreement no 801199; E.L.C. by a Rutherford Discovery Fellowship from the Royal Society of New Zealand Te Apārangi. Irish Whale and Dolphin Group Cetacean Stranding scheme is part-funded by the National Parks and Wildlife Service.

Acknowledgements. We thank L. Snyders, C. Luow and other representatives of the Khoisan Council of South Africa, B. Haami, Te Rūnanga o Ngāi Tahu of Aotearoa New Zealand, D. Neale, N. Scott and the New Zealand Department of Conservation Te Papa Atawhai (DOC) for advice, consultation and discussions on names for the new species; R. Stewart, Ngāti Māhaki and Te Rūnanga o Makaawhio, DOC, Museum of New Zealand Te Papa Tongarewa and the Haast community for assistance with the recovery and preparation of the type specimen, Nihongore; K. Andrews for ddRAD assistance; K. Murphy for help with extractions and library preparation; J. Klunk from Arbor Biosciences for facilitation capture and sequencing of NA *M. mirus* mitogenomes; R. Pitman, S. Cerchio, L. Conger and L. Hickmott, the NMFS survey team, NMFS, BOEM, the US Marine Mammal Commission, and partners for biopsy sample collection; P. Jepson, LNHM, J. Durban, L. Freitas, Madeira Whale Museum, L. Dollar, W. Perrin, M. Oremus, Marine Animal Stranding network of Northern Portugal and Fundo Ambiental, NZCeTA, and Southwest Fisheries Science Center's Marine Mammal and Sea Turtle Research Collection for sample access; J. Opperman and Iziko South African Museum for access to samples; M. Connan, Nelson Mandela University, for assistance in skull measurements, and W. Black and G. Alard for the pronunciation recording.

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
