## [Peer Review File · Proceedings of the Royal Society B: Biological Sciences]

Review History

RSPB-2021-1213.R0 (Original submission)

Review form: Reviewer 1

Recommendation

Accept with minor revision (please list in comments)

Scientific importance: Is the manuscript an original and important contribution to its field?

Good

General interest: Is the paper of sufficient general interest?

Good

Quality of the paper: Is the overall quality of the paper suitable?

Good

Is the length of the paper justified?

Yes

Should the paper be seen by a specialist statistical reviewer?

No

Do you have any concerns about statistical analyses in this paper? If so, please specify them explicitly in your report.

Yes

It is a condition of publication that authors make their supporting data, code and materials available - either as supplementary material or hosted in an external repository. Please rate, if applicable, the supporting data on the following criteria.

Is it accessible?

Yes

Is it clear?

Yes

Is it adequate?

Yes

Do you have any ethical concerns with this paper?

No

Comments to the Author

The taxonomy of the ziphiids has been challenging, and not always corresponding well between morphological and molecular classifications, in part due to the difficulty in assigning some animals found dead to species based on morphology. New species are being described with some regularity, and some species have been named based on very few data (including some never seen alive). In this context, the identification of new species is important to understanding the broader radiation. In general, this study provides compelling evidence that the southern and northern forms of True's beaked whale deserve separate species status. I have some comments and suggestions that I hope will be helpful in revision.

1) Line 71: Beaked whales don't actually inhabit the abyss, which technically starts at 4000m (e.g. Trujillo & Thurman, Essentials of Oceanography). Deepest known dive for a ziphiid (Cuvier's beaked whale) was just shy of 3000m, though of course routine dives are much shallower than that.

2) First paragraph of the intro is very similar to the abstract.

3) Line 149: would be quite useful to have a fuller description of the previous data on comparing True's beaked whales from these regions for both morphology and genetics. What were the morphological characters, how many samples, etc.?

4) Line 164: Not clear what the strategy was in choosing other species to include in the phylogenies. From earlier studies it seems that among the 14-15 Mesoplodon species *M. europaeus* and *M. ginglydens* cluster most closely to *M. mirus*, and these are included in the mitogenome tree. However, *M. bidens* and *M. densirostris* are used instead for the RAD tree, and these are not very close to *M. mirus* in earlier phylogenies. Explain the choice and strategy.

5) Line 178: 7X coverage is low for PSMC – some suggest a minimum of 20X. In the supplement you include the full 51X version of *M. mirus*, but without the bootstrapping. I think it would be good to show the bootstrapping, and perhaps other down-sampling levels, to give a better sense

of how much of an approximation/ distortion the 7X versions may be.

6) Lines 182-186: A series of bigger/ smaller type comparisons without any statistical support - whenever possible provide a statistical assessment.

7) Lines 190-191: Tajima's D values without statistical assessment - isn't meaningful unless significantly different from zero.

8) Line 217: Say if the holotype is included in the genetic analysis in this study, and point out in the figures if so.

9) Lines 250-252: With what evidence can you conclude that *M. eueu* is genetically differentiation from all other Mesoplodont species? You seem to have compared it against only 4 other species in this genus (out of ~15).

10) Line 254: Hard to be confident about fixed differences when the sample size is small (especially for the mitogenome where there are only 2 samples for *M. eueu*).

11) Lines 257-259: Comparisons of percent difference or fixed differences for ddRAD data should take into account the error rate for the method or at least mention it as a caveat. Error rate can potentially be quite high for RADseq methods (e.g. see Bresadola et al. 2020).

12) Line 281: interesting to know what the records are based on and how common (given the potential for misidentification for these species).

13) Line 306: didn't manage to find the details of how the mitogenome phylogeny was calibrated - were there fossil node calibrations? Could the confidence limits for the node dates calculated in Beast be presented, perhaps in a supplementary figure? Best guess is that they will be quite broad, which makes knowing when the species started to diverge difficult.

14) Line 351: Took me a while to locate sample sizes (in a supplementary table for mtDNA at least) - give numbers here for sample size by region for each analysis (citing supplementary information as appropriate).

15) Line 360: couldn't find clear reference to which samples were sequenced for ddRAD as part of this study, and for which species. As above, state in the text explicitly how many samples of each species and from where, how many samples per pool, sequenced in how many lanes.

16) Line 419: partitioning is described briefly for the mtDNA data (used partition finder, though more detail would be helpful), but there's no indication if any partitioning was tried for the ddRAD tree. Not straight-forward, but some do GC content etc. Probably not needed for this tree, given the small number of OTUs, but that could be acknowledged.

17) Line 479: say more about the strategy for removing PCR duplicates.

18) For morphological measurements, state the level of precision used, and how measurement error was quantified (typically done by replicate measurements). Also, explain briefly why a width measure was used for standardization rather than length, which is more conventional.

19) I find it useful to see the measurements illustrated in a figure, perhaps in the supplement?

Review form: Reviewer 2 (Travis Park)

Recommendation

Accept with minor revision (please list in comments)

Scientific importance: Is the manuscript an original and important contribution to its field?

Excellent

General interest: Is the paper of sufficient general interest?

Good

Quality of the paper: Is the overall quality of the paper suitable?

Good

Is the length of the paper justified?

Yes

Should the paper be seen by a specialist statistical reviewer?

No

Do you have any concerns about statistical analyses in this paper? If so, please specify them explicitly in your report.

No

It is a condition of publication that authors make their supporting data, code and materials available - either as supplementary material or hosted in an external repository. Please rate, if applicable, the supporting data on the following criteria.

Is it accessible?

Yes

Is it clear?

Yes

Is it adequate?

Yes

Do you have any ethical concerns with this paper?

No

Comments to the Author

This very solid paper establishes another new species of the beaked whale genus *Mesoplodon*, showcasing once again how much more we have to learn about our deep oceans. Whilst I am not an expert in molecular analyses, the results seem conclusive, and indicative of a species that has split from its sister taxon some time ago. The morphometrics results are not as stark, but they do show definite groupings in the morphospace. I think the more text could be added discussing the PCA, even if only to state how much variation the first two PCs comprise (see comments in PDF). An analysis using 3D geometric morphometrics would be nice, but is probably not worth the extra time and effort given it will only likely confirm these findings of the linear measurements as well as the conclusiveness of the molecular data. Another optional data thread could be comparisons of the inner ear which can be used to distinguish between species. I would also add a comparative image of the skull of *M. mirus* to figure 4 so that comparisons can be made by readers.

One other thing it would be interesting to see examined and discussed is whether or not there is any sexual dimorphism (other than body size) present in either species. Even if there is not, a comment should be made stating so.

This paper is also excellent in its explicit discussion of the involvement of the various indigenous groups in the naming of this paper. More studies should follow this example.

Overall, the paper is very well written. I have made my (minor) suggestions in the attached PDF and recommend publication when these corrections have been made. I look forward to seeing this paper in the literature. Well done to the authors. Travis Park.

Decision letter (RSPB-2021-1213.R0)

20-Aug-2021

Dear Dr Carroll:

Your manuscript has now been peer reviewed and the reviews have been assessed by an Associate Editor. The reviewers' comments (not including confidential comments to the Editor) and the comments from the Associate Editor are included at the end of this email for your reference. As you will see, the reviewers and the Editors have raised some concerns with your manuscript and we would like to invite you to revise your manuscript to address them.

Research ethics:

Use of animals and field studies:

It is a condition of publication that you make available the data and research materials supporting the results in the article. Please see our Data Sharing Policies (<https://royalsociety.org/journals/authors/author-guidelines/#data>). Datasets should be deposited in an appropriate publicly available repository and details of the associated accession number, link or DOI to the datasets must be included in the Data Accessibility section of the article (<https://royalsociety.org/journals/ethics-policies/data-sharing-mining/>). Reference(s) to datasets should also be included in the reference list of the article with DOIs (where available).

Please submit a copy of your revised paper within three weeks. If we do not hear from you within this time your manuscript will be rejected. If you are unable to meet this deadline please let us know as soon as possible, as we may be able to grant a short extension.

Best wishes,
Professor Gary Carvalho
<mailto:proceedingsb@royalsociety.org>

Associate Editor
Comments to Author:

Two expert reviewers agree that this study is well conducted, interesting, and of wider importance. Additionally, both reviewers present a series of well considered and specific points that should one addressed to make this study publishable. I agree with their recommendations.

Reviewer(s)' Comments to Author:

Referee: 1

Comments to the Author(s)

The taxonomy of the ziphiids has been challenging, and not always corresponding well between morphological and molecular classifications, in part due to the difficulty in assigning some animals found dead to species based on morphology. New species are being described with some regularity, and some species have been named based on very few data (including some never seen alive). In this context, the identification of new species is important to understanding the broader radiation. In general, this study provides compelling evidence that the southern and northern forms of True's beaked whale deserve separate species status. I have some comments and suggestions that I hope will be helpful in revision.

1) Line 71: Beaked whales don't actually inhabit the abyss, which technically starts at 4000m (e.g. Trujillo & Thurman, Essentials of Oceanography). Deepest known dive for a ziphiid (Cuvier's beaked whale) was just shy of 3000m, though of course routine dives are much shallower than that.

2) First paragraph of the intro is very similar to the abstract.

3) Line 149: would be quite useful to have a fuller description of the previous data on comparing True's beaked whales from these regions for both morphology and genetics. What were the morphological characters, how many samples, etc.?

4) Line 164: Not clear what the strategy was in choosing other species to include in the phylogenies. From earlier studies it seems that among the 14-15 *Mesoplodon* species *M. europaeus* and *M. gingodens* cluster most closely to *M. mirus*, and these are included in the mitogenome tree. However, *M. bidens* and *M. densirostris* are used instead for the RAD tree, and these are not very close to *M. mirus* in earlier phylogenies. Explain the choice and strategy.

5) Line 178: 7X coverage is low for PSMC – some suggest a minimum of 20X. In the supplement you include the full 51X version of *M. mirus*, but without the bootstrapping. I think it would be good to show the bootstrapping, and perhaps other down-sampling levels, to give a better sense of how much of an approximation/ distortion the 7X versions may be.

6) Lines 182-186: A series of bigger/ smaller type comparisons without any statistical support - whenever possible provide a statistical assessment.

7) Lines 190-191: Tajima's D values without statistical assessment – isn't meaningful unless significantly different from zero.

8) Line 217: Say if the holotype is included in the genetic analysis in this study, and point out in the figures if so.

9) Lines 250-252: With what evidence can you conclude that *M. eueu* is genetically differentiation from all other *Mesoplodont* species? You seem to have compared it against only 4 other species in this genus (out of ~15).

10) Line 254: Hard to be confident about fixed differences when the sample size is small (especially for the mitogenome where there are only 2 samples for *M. eueu*).

- 11) Lines 257-259: Comparisons of percent difference or fixed differences for ddRAD data should take into account the error rate for the method or at least mention it as a caveat. Error rate can potentially be quite high for RADseq methods (e.g. see Bresadola et al. 2020).
- 12) Line 281: interesting to know what the records are based on and how common (given the potential for misidentification for these species).
- 13) Line 306: didn't manage to find the details of how the mitogenome phylogeny was calibrated – were there fossil node calibrations? Could the confidence limits for the node dates calculated in Beast be presented, perhaps in a supplementary figure? Best guess is that they will be quite broad, which makes knowing when the species started to diverge difficult.
- 14) Line 351: Took me a while to locate sample sizes (in a supplementary table for mtDNA at least) – give numbers here for sample size by region for each analysis (citing supplementary information as appropriate).
- 15) Line 360: couldn't find clear reference to which samples were sequenced for ddRAD as part of this study, and for which species. As above, state in the text explicitly how many samples of each species and from where, how many samples per pool, sequenced in how many lanes.
- 16) Line 419: partitioning is described briefly for the mtDNA data (used partition finder, though more detail would be helpful), but there's no indication if any partitioning was tried for the ddRAD tree. Not straight-forward, but some do GC content etc. Probably not needed for this tree, given the small number of OTUs, but that could be acknowledged.
- 17) Line 479: say more about the strategy for removing PCR duplicates.
- 18) For morphological measurements, state the level of precision used, and how measurement error was quantified (typically done by replicate measurements). Also, explain briefly why a width measure was used for standardization rather than length, which is more conventional.
- 19) I find it useful to see the measurements illustrated in a figure, perhaps in the supplement?

Referee: 2

Comments to the Author(s)

This very solid paper establishes another new species of the beaked whale genus *Mesoplodon*, showcasing once again how much more we have to learn about our deep oceans. Whilst I am not an expert in molecular analyses, the results seem conclusive, and indicative of a species that has split from its sister taxon some time ago. The morphometrics results are not as stark, but they do show definite groupings in the morphospace. I think the more text could be added discussing the PCA, even if only to state how much variation the first two PCs comprise (see comments in PDF). An analysis using 3D geometric morphometrics would be nice, but is probably not worth the extra time and effort given it will only likely confirm these findings of the linear measurements as well as the conclusiveness of the molecular data. Another optional data thread could be comparisons of the inner ear which can be used to distinguish between species. I would also add a comparative image of the skull of *M. mirus* to figure 4 so that comparisons can be made by readers.

One other thing it would be interesting to see examined and discussed is whether or not there is any sexual dimorphism (other than body size) present in either species. Even if there is not, a comment should be made stating so.

This paper is also excellent in its explicit discussion of the involvement of the various indigenous groups in the naming of this paper. More studies should follow this example.

Overall, the paper is very well written. I have made my (minor) suggestions in the attached PDF and recommend publication when these corrections have been made. I look forward to seeing this paper in the literature. Well done to the authors. Travis Park.

Author's Response to Decision Letter for (RSPB-2021-1213.R0)

See Appendix A.

Decision letter (RSPB-2021-1213.R1)

16-Sep-2021

Dear Dr Carroll

I am pleased to inform you that your manuscript RSPB-2021-1213.R1 entitled "Speciation in the deep - genomics and morphology reveal a new species of beaked whale &emdash;Mesoplodon eueu&emdash;" has been accepted for publication in Proceedings B.

Please can you provide a link to your genomic data that has been deposited in a repository in the Data Accessibility section of our submission system.

Because the schedule for publication is very tight, it is a condition of publication that you submit the revised version of your manuscript within 7 days. If you do not think you will be able to meet this date please let us know.

4) If you wish to submit your data to Dryad (<http://datadryad.org/>) and have not already done so you can submit your data via this link [http://datadryad.org/submit?journalID=RSPB&manu=\(Document not available\)](http://datadryad.org/submit?journalID=RSPB&manu=(Document%20not%20available)) which will take you to your unique entry in the Dryad repository. If you have already submitted your data to dryad you can make any necessary revisions to your dataset by following the above link. Please see <https://royalsociety.org/journals/ethics-policies/data-sharing-mining/> for more details.

Sincerely,
Professor Gary Carvalho
Editor, Proceedings B
<mailto:proceedingsb@royalsociety.org>

Associate Editor:

Board Member

Comments to Author:

The authors have provided convincing responses to all the reviewer comments (which were minor) and the ms includes appropriate clarifications and extra details, as requested.

Decision letter (RSPB-2021-1213.R2)

30-Sep-2021

Dear Dr Carroll

I am pleased to inform you that your manuscript entitled "Speciation in the deep - genomics and morphology reveal a new species of beaked whale *Mesoplodon eueu*" has been accepted for publication in Proceedings B.

Data Accessibility section

Open Access

Paper charges

Sincerely,

Proceedings B

Appendix A

Emma Carroll PhD
Rutherford Discovery Fellow
School of Biological Sciences
University of Auckland

Thomas Building
Building 110
3a Symonds Street
Auckland Central 1010

e: e.carroll@auckland.ac.nz
p: +64 9 3737 599 x 88239
t: EmzL.Carroll
w: emmacarrollphd.wordpress.com

10 September 2021

Re: Submission of 'Speciation in the abyss - genomics and morphology reveal a new species of beaked whale'

Dear Editor,

Thank you for the opportunity to revise and resubmit our manuscript. We appreciated the Reviewer's helpful suggestions and requests to clarify our work, but also their recognition of the comprehensiveness of our analyse, spanning both genomic and morphological work. We also appreciated the recognition of our engagement with Indigenous peoples in the naming of the new species. We agree that this is an important part of the decolonisation of science and are excited to be part of the broadening of the conversation around naming new species, particularly charismatic megafauna like whales.

As you will see, we have undertaken new analyses to address the Reviewer's comments, including new morphometric analyses to assess sex differences within species, and statistical tests to assess whether the new Southern Hemisphere species does indeed have higher levels of genetic diversity.

Furthermore, we have clarified details on sample size and provide more information in supplementary datasets on sample metadata, for both the genomic and morphometric data.

Additionally, we provide an audioclip of the Khwe word from which the new species name is derived. As the word has to be latinised to become part of the species' scientific name, the accents that help define the click in the word are lost. Having an audio file, as suggested by a Reviewer, will help ensure that the correct pronunciation is adopted from the outset.

I look forward to hearing from you.

Kind regards

Emma Carroll

Response to Reviewers: Speciation in the deep sea - genomics and morphology reveal a new species of beaked whale RSPB-2021-1213
Please find the original comments in standard text and our responses below in **bold**.

Dear Dr Carroll:

Your manuscript has now been peer reviewed and the reviews have been assessed by an Associate Editor. The reviewers' comments (not including confidential comments to the Editor) and the comments from the Associate Editor are included at the end of this email for your reference. As you will see, the reviewers and the Editors have raised some concerns with your manuscript and we would like to invite you to revise your manuscript to address them.

Please submit a copy of your revised paper within three weeks. If we do not hear from you within this time your manuscript will be rejected. If you are unable to meet this deadline please let us know as soon as possible, as we may be able to grant a short extension.

Best wishes,

Professor Gary Carvalho
mailto: proceedingsb@royalsociety.org

Associate Editor

Comments to Author:

Two expert reviewers agree that this study is well conducted, interesting, and of wider importance. Additionally, both reviewers present a series of well considered and specific points that should one addressed to make this study publishable. I agree with their recommendations.

Reviewer(s)' Comments to Author:

Referee: 1

Comments to the Author(s)

The taxonomy of the ziphiids has been challenging, and not always corresponding well between morphological and molecular classifications, in part due to the difficulty in assigning some animals found dead to species based on morphology. New species are being described with some regularity, and some species have been named based on very few data (including some never seen alive). In this context, the identification of new species is important to understanding the broader radiation. In general, this study provides compelling evidence that the southern and northern forms of True's beaked whale deserve separate species status. I have some comments and suggestions that I hope will be helpful in revision.

Reviewer 1 comment 1) Line 71: Beaked whales don't actually inhabit the abyss, which technically starts at 4000m (e.g. Trujillo & Thurman, Essentials of Oceanography). Deepest known dive for a ziphiid (Cuvier's beaked whale) was just shy of 3000m, though of course routine dives are much shallower than that. **The Reviewer is technically correct, although 'abyss' has more colloquial uses on which we were drawing. We have replaced the use of 'abyss' in the title and text with 'deep' or 'deep sea'.**

Reviewer 1 comment 2) First paragraph of the intro is very similar to the abstract. **We have edited the abstract and paragraph to reduce the similarities (see tracked changes version pages 3 and 4).**

Reviewer 1 comment 3) Line 149: would be quite useful to have a fuller description of the previous data on comparing True's beaked whales from these regions for both morphology and genetics. What were the morphological characters, how many samples, etc.? **The previous work to date has been limited, with the only systematic morphological assessment done by Ross (1984) based on six samples from South Africa and nine from the North Atlantic. This found one of 47 cranial measurements distinguished the regions, which led him to conclude they were definitely the same species. Molecular analyses has similarly been limited; four samples were analysed by Dalebout et al (2007, 2014) as part of other studies, and the divergence between hemispheres was noted. A few studies using genetic data to confirm species ID have also noted and confirmed differentiation between hemispheres, but in an ad hoc way (e.g., Aguilar et al 2017). This is now outlined in more detail in the Introduction (Line 165 onwards).**

Reviewer 1 comment 4) Line 164: Not clear what the strategy was in choosing other species to include in the phylogenies. From earlier studies it seems that among the 14-15 Mesoplodon species *M. europaeus* and *M. ginglydens* cluster most closely to *M. mirus*, and these are included in the mitogenome tree. However, *M. bidens* and *M. densirostris* are used instead for the RAD tree, and these are not very close to *M. mirus* in earlier phylogenies. Explain the choice and strategy.

There were different strategies behind the selection of samples for the ddRAD and mitochondrial based analyses. For ddRAD our strategy was outlined in the Methods: we used a dataset comprising the species of interest, *M. mirus*, and two other globally distributed beaked whales to compare divergence patterns. Furthermore, we included *M. bidens*, a more closely related species to *M. mirus*, which is distributed in the North Atlantic only, comparable to the revised *M. mirus*. This has been clarified in the Results as it precedes the Methods (line 191 onwards).

For the mitogenome analysis, we specifically selected the focal species and its previously identified closest relatives, as highlighted by the reviewer. This is more clearly stated in the Methods and Results (lines 206 and 763).

Reviewer 1 comment 5) Line 178: 7X coverage is low for PSMC – some suggest a minimum of 20X. In the supplement you include the full 51X version of *M. mirus*, but without the bootstrapping. I think it would be good to show the bootstrapping, and perhaps other down-sampling levels, to give a better sense of how much of an approximation/ distortion the 7X versions may be.

Following this comment, we have explored the impact of downsampling across a range of values: 5x, 7x, 10x, 20x, 30x, and 40x, and present the median values and their bootstrap replicate distributions in Figure S1. What we found essentially the same pattern as described in the original manuscript; the pattern of changes through time is broadly similar, but the downsampled PSMC results show a flattened trend compared with higher coverage.

Reviewer 1 comment 6) Lines 182-186: A series of bigger/ smaller type comparisons without any statistical support - whenever possible provide a statistical assessment.

We apologise for presenting and comparing these estimates without variance or statistical support. We have revised the paper to present these statistics with 95% CI and tested for significant differences. The nuclear DNA results show that SH has statistically significant higher genetic diversity but the two species do not show statistical differences in the mtDNA haplotype diversity measures. See Methods and Results revisions (lines 214, 616, 702, 720).

Reviewer 1 comment 7) Lines 190-191: Tajima's D values without statistical assessment – isn't meaningful unless significantly different from zero.

Thank you for highlight this. The Tajima's D results have been updated with to include variance estimates and results of tests that indicate the mean of the distribution of Tajima's D values is significantly different from zero (t-test and Wilcoxon tests, $p < 0.001$). See Methods and Results revisions (lines 225, 614).

Reviewer 1 comment 8) Line 217: Say if the holotype is included in the genetic analysis in this study, and point out in the figures if so.

The holotypes for both *M. mirus* and *M. eueu* are included in the mitochondrial phylogeny, and the ddRAD analysis for the proposed *M. eueu* holotype (as shown on Figure 2 by stars and noted in the Figure legend.) The use of holotypes in both the genetic and morphological analyses is more clearly explained in the manuscript and species description. (e.g., lines 184, 231).

Reviewer 1 comment 9) Lines 250-252: With what evidence can you conclude that *M. eueu* is genetically differentiated from all other Mesoplodont species? You seem to have compared it against only 4 other species in this genus (out of ~15).

This is partly based on published work as *M. mirus* and *M. eueu* have been confirmed to be different to all other mesoplodont species based on previous control region phylogeny (e.g., Dalebout et al 2007) and *M. mirus* is confirmed to be different from other mesoplodonts based on substantial nuclear data (McGowen et al., 2020).

We have edited this section to read (line 313): *M. eueu* differs from *M. mirus* based on nuclear DNA markers and from *M. mirus* and its closest relatives *M. europaeus*, *M. ginkgodens* and *M. bidens* using mtDNA markers (figure 2). *M. mirus* is distinct from all other mesoplodont species based on previous mitochondrial and nuclear DNA trees [7,20,23].

Reviewer 1 comment 10) Line 254: Hard to be confident about fixed differences when the sample size is small (especially for the mitogenome where there are only 2 samples for *M. eueu*).

We acknowledge that this statistic is limited by sample size, and the also reported F_{ST} and d_A statistics are more meaningful indicators of differentiation given the available data. However, it is convention for the number of fixed differences to be stated in species description for cetaceans, despite small sample sizes. In fact for this taxa, we have a larger number of specimens analysed for the short mtDNA fragment (14 for *M. eueu* and 19 for *M. mirus*) compared with previous published species descriptions (e.g., 5 for Dalebout et al 2014; 2-8 per species for *Berardius* spp, Yamada et al 2018) that similarly state the number of fixed differences.

Reviewer 1 comment 11) Lines 257-259: Comparisons of percent difference or fixed differences for ddRAD data should take into account the error rate for the method or at least mention it as a caveat. Error rate can potentially be quite high for RADseq methods (e.g. see Bresadola et al. 2020).

One replicate sample was used to estimate error and identify and remove discordant loci in the ddRAD genotype calling pipeline. This involved one sample being processed through the whole lab and genotype calling process twice, and based on this, the error rate was 0.002 per SNP allele prior to discordant loci being removed. This means that the error rate is likely lower in the final dataset as loci showing error in this duplicate were removed from all samples, but this indicative error rate is about one order of magnitude lower than the estimated Illumina sequencing error rate

(<https://www.nature.com/articles/s41598-018-29325-6>), giving confidence in our filtering and QC process. This error rate is now mentioned in the manuscript (see lines 199, 325, 598) and species description. However, these statistics report fixed differences between species, not just variable sites, and a detailed description about how genotyping error could inflate or deflate the number of fixed differences observed here is beyond the scope of the current manuscript and limited by space.

Reviewer 1 comment 12) Line 281: interesting to know what the records are based on and how common (given the potential for misidentification for these species). **This is a good question and so we have given more details on where the species identification has been confirmed with published genetic data; in this paper and previously (line 350).**

Reviewer 1 comment 13) Line 306: didn't manage to find the details of how the mitogenome phylogeny was calibrated – were there fossil node calibrations? Could the confidence limits for the node dates calculated in Beast be presented, perhaps in a supplementary figure? Best guess is that they will be quite broad, which makes knowing when the species started to diverge difficult. **There were indeed calibrations based on the results of McGowen et al. (2020), the details for which are presented in table S4 (including confidence limits). The analyses of McGowen et al. (2020) used extensive fossil calibrations in their analysis of the complete cetacean tree. We have edited the text in the main manuscript and supplementary table to make this clearer (line 685, table S4).**

Reviewer 1 comment 14) Line 351: Took me a while to locate sample sizes (in a supplementary table for mtDNA at least) – give numbers here for sample size by region for each analysis (citing supplementary information as appropriate). **For the original submission, all of these data were available in the Supplementary Dataset, which also included additional metadata such as collection date, sex, etc. This may not have been available to the Reviewer. Sample sizes are also indicated by in the results paragraph at the start of the manuscript (e.g., lines 182, 190, paragraph line 197 where n = is given).**

In response to this comment we have created a 'Dataset Summary' section of the Methods, which clearly states where such information are available. We have also included table S1, which is a summary of the which samples were used in which analyses, summarised by species and geographic location, with full details on sample metadata available in dataset S1 for genomics and dataset S2 for morphology (see line 509).

Reviewer 1 comment 15) Line 360: couldn't find clear reference to which samples were sequenced for ddRAD as part of this study, and for which species. As above, state in the text explicitly how many samples of each species and from where, how many samples per pool, sequenced in how many lanes. **See response above. Additional details have been given in the Methods section as well, but the citation (Carroll et al 2016) is freely available and has detailed information on the protocol that allows full replication of the work, which could not be reproduced here due to space limitations.**

Reviewer 1 comment 16) Line 419: partitioning is described briefly for the mtDNA data (used partition finder, though more detail would be helpful), but there's no indication if any partitioning was tried for the ddRAD tree. Not straight-forward, but some do GC content etc. Probably not needed for this tree, given the small number of OTUs, but that could be acknowledged. **No partitioning was used, this has been clarified in the methods (line 608).**

Reviewer 1 comment 17) Line 479: say more about the strategy for removing PCR duplicates.

PCR duplicates were removed using the rmdup command in Samtools. This has been clarified (line 672).

Reviewer 1 comment 18) For morphological measurements, state the level of precision used, and how measurement error was quantified (typically done by replicate measurements). Also, explain briefly why a width measure was used for standardization rather than length, which is more conventional.

To address this comment we have added the following text; “Measurements were taken to the nearest mm, and all measurements were divided by the bizygomatic width of the skull to account for differences in body size (see [76] for the use of this metric as a size proxy in cetaceans)” (line 755).

We do not think it is necessary to report measurement error in this case, as the five morphometric measurements that most strongly differentiate the species show mean differences of between 14 and 39 mm, representing ~10% of the original mean measurement value in each case. Measurement error is generally a problem when true phenotypic variation is small (Arnqvist & Martensson 1998).

It is also not possible to assess this due to lockdowns and the difficulty in getting access to the samples; we find it somewhat miraculous that these data were compiled and analysed over the past 12 months as it is. Repeated measurements are currently hindered by an ongoing lockdown in New Zealand, for example.

Reviewer 1 comment 19) I find it useful to see the measurements illustrated in a figure, perhaps in the supplement?

This has been provided in figure S4. Thank you for the suggestion.

Referee: 2

Comments to the Author(s)

This very solid paper establishes another new species of the beaked whale genus *Mesoplodon*, showcasing once again how much more we have to learn about our deep oceans. Whilst I am not an expert in molecular analyses, the results seem conclusive, and indicative of a species that has split from its sister taxon some time ago. The morphometrics results are not as stark, but they do show definite groupings in the morphospace.

We thank the review for their kind comments.

Reviewer 2 comment 1: I think the more text could be added discussing the PCA, even if only to state how much variation the first two PCs comprise (see comments in PDF).

In response to this comment and others in the PDF annotated by the Reviewer, we have:

- **Provided full details of all data used in morphometrics in dataset S2, including sample measurements and metadata, transformed measurements used in the PCAs, results of PCAs (eigenvalue and % variation for each PC and PC value for each sample) for comparison of SH and NA, as well as sexes within each hemisphere**
- **Provided the % variation explained by the first two PCs and that 6 PCs are required to explain 95% of the variation in the data in the text (see paragraph line 230)**
- **Provided the first two PC loadings for the NA/SH comparison in figure S5**
- **Provided the PCA scatter plot and first two principal component loadings comparing males and females for *M. eueu* and *M. mirus* in figure S6.**

Reviewer 2 comment 2: An analysis using 3D geometric morphometrics would be nice, but is probably not worth the extra time and effort given it will only likely confirm these findings of the linear measurements as well as the conclusiveness of the molecular data. Another optional data thread could be comparisons of the inner ear which can be used to distinguish between species. I would also add a comparative image of the skull of *M. mirus* to figure 4 so that comparisons can be made by readers.

We agree with the reviewer that the use of 3D geometric morphometrics, and inclusion of additional morphological features such as the inner ear, are excellent next steps to analyse variation between and within these species. However, given the strength of the molecular data and the findings of the existing morphological data highlighted by the reviewer, and the difficulty in doing additional lab based work in a pandemic characterised lockdowns, it is beyond the scope of the current paper. We have also added a comparative image of *M. mirus* (female: figure S2; male: figure S8) and a male *M. eueu* (figure S3) to the supplementary material, given that we are short on space and figure 4 has several components to it already.

Reviewer 2 comment 3: One other thing it would be interesting to see examined and discussed is whether or not there is any sexual dimorphism (other than body size) present in either species. Even if there is not, a comment should be made stating so. **We agree and thank the Reviewer for the suggestion. Please see response to Reviewer 1 comment 2.**

Reviewer 2 comment 4: This paper is also excellent in its explicit discussion of the involvement of the various indigenous groups in the naming of this paper. More studies should follow this example. **We thank the reviewer for their positive comment and agree that inclusion of Indigenous communities should be more widespread in species naming initiatives.**

Reviewer 2 comment 5: Overall, the paper is very well written. I have made my (minor) suggestions in the attached PDF and recommend publication when these corrections have been made. I look forward to seeing this paper in the literature. Well done to the authors. Travis Park.

We thank the Reviewer for the positive comments and have incorporated these edits.

Additional Reviewer 2 comments/questions from this pdf:

Reviewer 2 comment 6: Is there any evidence of current inbreeding/hybridism between the two taxa?

The ddRAD analyses clearly cluster the two taxa in to distinct clades, and the sNMF genetic clustering analysis did not show any signs of introgression. A sentence to the effect has been added to the ms (page xxx line yyy).

Reviewer 2 comment 7: Please also list how many PCs it takes to comprise 95% of the variation. And add a table listing the PC values in the ESM would be useful.

Please see response to Reviewer 2 comment 1.

Reviewer 2 comment 8: Add specimen number of skull to fig caption

This has been added to figure 2.

Reviewer 2 comment 9: a description of how to say the species name

An audio clip of the correct pronunciation has been uploaded to be linked with the manuscript; see the cleaned ms with [h] added where we propose it is hyperlinked to in the published paper.